# RECURRENT MIXTURE DENSITY NETWORK FOR SPATIOTEMPORAL VISUAL ATTENTION

**Loris Bazzani**[*]
Amazon
Berlin, Germany
bazzanil@amazon.com

**Hugo Larochelle**[†]
Département d'informatique
Université de Sherbrooke
hugo.larochelle@usherbrooke.ca

**Lorenzo Torresani**
Department of Computer Science
Dartmouth College
lt@dartmouth.edu

## ABSTRACT

In many computer vision tasks, the relevant information to solve the problem at hand is mixed with irrelevant, distracting information. This has motivated researchers to design *attentional models* that can dynamically focus on parts of images or videos that are salient, *e.g.*, by down-weighting irrelevant pixels. In this work, we propose a spatiotemporal attentional model that learns *where to look* in a video directly from human fixation data. We model visual attention with a mixture of Gaussians at each frame. This distribution is used to express the probability of saliency for each pixel. Time consistency in videos is modeled hierarchically by: 1) deep 3D convolutional features to represent spatial and short-term time relations at clip level and 2) a long short-term memory network on top that aggregates the clip-level representation of sequential clips and therefore expands the temporal domain from few frames to seconds. The parameters of the proposed model are optimized via maximum likelihood estimation using human fixations as training data, without knowledge of the action in each video. Our experiments on Hollywood2 show state-of-the-art performance on saliency prediction for video. We also show that our attentional model trained on Hollywood2 generalizes well to UCF101 and it can be leveraged to improve action classification accuracy on both datasets.

## 1 INTRODUCTION

Attentional modeling and saliency prediction in images has been an active research topic in computer vision over the last decade. Interest in attentional models is primarily motivated by their ability to eliminate or down-weight pixels that are not important for the task at hand, as for example shown in prior work using visual attention for image recognition and caption generation (Sermanet et al., 2014; Xu et al., 2015; Mnih et al., 2014). Integrating visual attention in an image analysis model can potentially lead to improved overall accuracy, as the system can focus on the most salient regions in the photo without being disturbed by irrelevant information.

Recently, we have witnessed a shift of trend from image saliency prediction (Borji & Itti, 2013) to the modeling of saliency in videos (Rudoy et al., 2013). Since human fixation patterns are strongly correlated over time (Coull, 2004), it appears critical to model the relations between saliency maps of consecutive frames. In this scenario, attention can be defined as a spatiotemporal volume, where each saliency map (one for each frame) depends on the frames at the previous times. The saliency map can be interpreted as a probability distribution over pixels and the actual fixation patterns can be generated by sampling from the the map.

---

[*]This work was done when Loris Bazzani was at Dartmouth College.
[†]Hugo Larochelle is now at Google Brain.

Going from images to videos is not straightforward, since videos bring up many challenges. First of all, videos have an additional dimension (time), compared to images. This causes a dramatic growth in the number of pixels to be processed and poses a significantly higher computational cost for analysis. At the same time, there are strong redundancies present in such data, which implies that visual attention may be particularly beneficial for the video setting. For example, typically the objects or people in a video do not change significantly in appearance over time. Yet, for analysis tasks such as action recognition (Wang & Schmid, 2013) or video description (Yao et al., 2015), it is imperative to properly model the dynamical properties of these objects in the video. This suggests that, in order to identify spatiotemporal volumes that are salient for video analysis, an attentional model must take into account high-level image semantics as well as the history of past fixations.

In order to cope with these challenges, we propose an efficient spatiotemporal attentional model (see Fig. 1) that leverages deep 3D convolutional features (Tran et al., 2015) as semantic, spatiotemporal representation of short clips in the video. This clip-level representation is then aggregated by a Long Short-Term Memory (LSTM) network (Hochreiter & Schmidhuber, 1997), that expands the temporal range of analysis from few frames to seconds. The LSTM model connects into a Mixture Density Network (MDN) (Bishop, 1994) that at each frame outputs the parameters of a Gaussian mixture model expressing the saliency map. We refer to this model as Recurrent Mixture Density Network (RMDN). RMDN is trained via maximum likelihood estimation using human fixations as training data, without knowledge of the actions in the videos.

The potential applications of automatic saliency map prediction from videos are many. They include attention-based video compression (Gitman et al., 2014), visual attention for robots (Yu et al., 2010), crowd analysis for video surveillance (Jiang et al., 2014), salient object detection (Li & Yu, 2015; Karthikeyan et al., 2015) and activity recognition (Vig et al., 2012; Sapienza et al., 2014). In this work we focus on a study of how visual attention may improve action recognition by leveraging the saliency map generated by RMDN for video classification. The idea is akin to soft attention and consists in re-weighting the pixel values of the input video by the estimated saliency map. Despite its simplicity, we show that the combination of features extracted from this modified version of the video and those computed from the original input lead to a significant improvement in action recognition, compared to a model that does not use attention.

The primary contribution of this work is a spatiotemporal saliency estimation network optimized to reproduce human fixations. The proposed approach offers several advantages: 1) the model can be trained without having to engineer spatiotemporal features; 2) RMDN is directly trained on examples of human fixations and thus learns to mimic human visual attention; 3) prediction of the saliency map is very fast (it takes $0.08$s per 16-frame clip on a GPU); 4) the method outperforms the state-of-the-art (Mathe & Sminchisescu, 2015) in saliency accuracy; 5) our predicted saliency maps lead to to improvements in action classification accuracy.

## 2 RELATED WORK

Broadly speaking, the literature on attentional models can be split into two categories: *task-agnostic* approaches which model the bottom-up, free-viewing properties of attention, and *task-specific* methods which model its top-down, task-driven properties. Researchers have devoted many years to create datasets, collecting human fixations and proposing solutions for biologically-plausible saliency estimators, built using low-level cues such as edge detectors and color filters (e.g. see Borji et al. (2013); Judd et al. (2009); Harel et al. (2006) for recent examples). We refer to Borji & Itti (2013) and Bruce et al. (2016) for an interesting analysis and comparison of existing methods. Most of the techniques in the literature are focused on extracting features in a bottom-up and/or top-down manner and use them to estimate the saliency map. In this context, motion features are introduced when extending saliency methods from images to videos (Guo et al., 2008; Zhao et al., 2015; Zhai & Shah, 2006). However, there is no explicit modeling of the temporal dimension that can capture long-term relations. In fact, motion features (*e.g.*, optical flow) describe short-term associations at the temporal scale of only a few consecutive frames.

Prior approaches can also be categorized into *soft-attentional* versus *hard-attentional* models. Soft-attentional models use the predicted saliency maps to down-weight pixels that are not relevant or salient, *e.g.*, Song et al. (2016). Specifically deep networks have been used in this context to assign a weight to each pixel in order to extract "glimpses" from images (Xu et al., 2015; Gregor et al., 2015)

or videos (Yao et al., 2015) in the form of weighted pixel averages. One strength of such approaches is that they can backpropagate through the attentional component and tune it in the context of its use in a deep network. Other work has been geared towards learning hard-attentional models, which explicitly ignore and discard parts of the input (Larochelle & Hinton, 2010; Bazzani et al., 2011; Denil et al., 2012; Mnih et al., 2014; Xu et al., 2015; Sermanet et al., 2014; Ba et al., 2015; Yoo et al., 2015; Zheng et al., 2015), thus providing significant computational savings. Unfortunately, such models are often hard to train because they rely on reinforcement learning techniques to generate the image/video locations during training.

All of the aforementioned prior work attempts to learn attentional models indirectly rather than from explicit information of where humans look. Recent work (Mauthner et al., 2015; Hossein Khatoon-abadi et al., 2015; Mathe & Sminchisescu, 2015; Stefan Mathe, 2013; Kümmerer et al., 2015; Rudoy et al., 2013) has shown that it may be possible to accurately reproduce gazing patterns of human subjects attending to images and videos. However, these prior approaches rely on hand-crafted features to estimate the saliency maps. Attempts at removing hand-engineering of features are represented by Jetley et al. (2016); Huang et al. (2015); Kümmerer et al. (2015) where networks pre-trained for object recognition were subsequently finetuned using saliency-based loss functions for images. Pan & i Nieto (2015) followed the same principle but without using any pre-trained network for initialization. Liu et al. (2015) proposed a multi-scale architecture for saliency prediction, and Li & Yu (2015) added a refinement step in order to enforce spatial coherence of the output. Simonyan et al. (2014) and Mahendran & Vedaldi (2016) proposed to reverse deep networks using deconvolutions for visualization and to estimate image saliency. However, these methods estimate saliency from still images and do not consider the temporal aspect of video. Chaabouni et al. (2016a;b) trained a ConvNet for saliency prediction on optical flow features and individual frames. However the model uses only the very short-term temporal relations of two consecutive frames.

In this paper, we explore the following question: can deep networks be trained to reliably predict spatiotemporal attentional patterns, specifically in such a way that these predictions can be leveraged successfully by a recognition system? To our knowledge, our work distinguishes itself from the aforementioned literature by being the first application of deep networks to the prediction of spatiotemporal human saliency in videos.

## 3 PROPOSED MODEL

We start with a high-level description of our attentional model. We then formalize it in Sec. 3.1, and describe its training in Sec. 3.2. Sec. 3.3 reports how prediction is efficiently carried out at test time. Sec. 3.4 describes how to leverage the predicted saliency map to improve action recognition.

The proposed RMDN model for saliency estimation is depicted in Fig. 1. At time $t$, the input of the model is a sequence of the last $K = 16$ frames, *i.e.*, from time $t-K+1$ to current time $t$. We refer to this sequence as the input "clip." The first part of the model (Fig. 1, blue layers above the input clip) consists of a 3D convolutional network that provides a feature representation of the clip. Our choice of a clip-based representation rather than a single-frame descriptor is motivated by the fact that these features allow us to explicitly capture short-term information that is then aggregated for long-term spatiotemporal visual attention by RMDN. Furthermore, there is recently growing evidence (Tran et al., 2015; Srivastava et al., 2015; Yue-Hei Ng et al., 2015) that by modeling the temporal information it is possible to obtain improved performances in high-level video analysis tasks, such as action recognition. For the computation of spatiotemporal features from the input clip, we use the "C3D" architecture proposed by Tran et al. (2015), which has been shown to provide competitive results on video recognition tasks across different datasets. The C3D architecture is defined as: C64-P-C128-P-C256-C256-P-C512-C512-P-C512-C512-P-FC4096-FC4096-softmax, where C is a 3D convolutional layer, P is the pooling layer, FC is a fully-connected layer, and the number specifies the number of kernels of the layer (e.g. C64 has 64 kernels). For the details about the size and stride of the convolutional and pooling kernels, we refer to (Tran et al., 2015).

The convolutional network has access to a limited window of the video since it uses a fixed-size clip of 16 frames as input. In order to empower the visual attention model with the ability to take into account longer temporal extents, we need a mechanism that performs temporal aggregation of past clip-level signals. To this end, we propose to connect the internal representation of the C3D model to a recurrent neural network, as shown in Fig. 1 (green module). The aim of the temporal

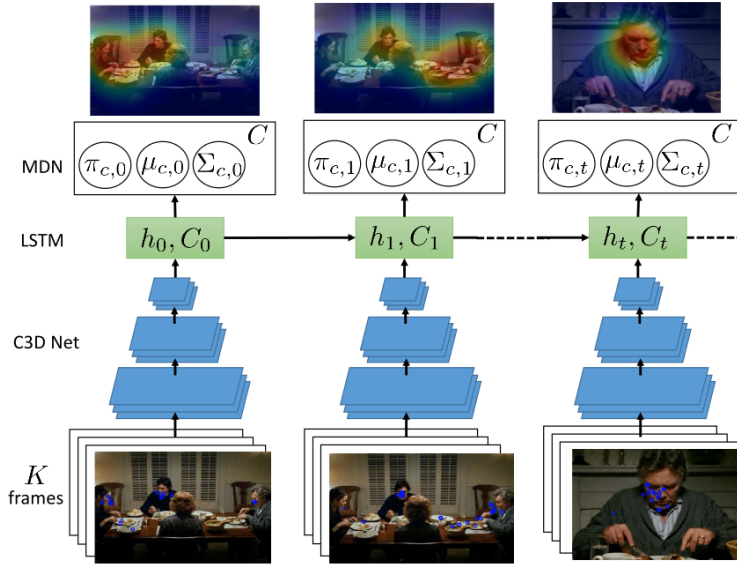

Figure 1: Proposed recurrent mixture density network for saliency prediction. The input clip of $K$ frames is fed into a 3D convolutional network (in blue), whose output becomes the input of a long short-term memory (LSTM) network (in green). Finally, a linear layer projects the LSTM representation to the parameters of a Gaussian mixture model, which describes the saliency map.

connections of the recurrent neural network is to propagate the clip-level features through time via memory units that can capture long-term dependencies. Our model uses LSTMs (Hochreiter & Schmidhuber, 1997) as memory blocks.

The saliency map at each time $t$ is expressed in terms of a Gaussian Mixture Model (GMM) with $C$ components. We denote its parameters with $\{(\mu^c, \pi^c, \Sigma^c)\}_{c=1}^C$, where $\mu^c$, $\pi^c$ and $\Sigma^c$ are the mean, the mixture coefficient and the covariance of the $c$-th Gaussian component, respectively. The LSTM directly outputs these parameters (see details below). The resulting network is known as a Mixture Density Network (MDN) (Bishop, 1994; Graves, 2013).

Since the model is recurrent, there is a direct connection between the inner representation of the LSTM at time $t$ and the one at time $t + 1$. This favors temporal consistency between the saliency maps at adjacent times.

## 3.1 FORMALIZATION OF THE MODEL

Let $\mathcal{D} = \{(\mathbf{v}^i, \mathbf{a}^i)\}_{i=1}^N$ be a dataset of videos and human fixations pairs. $\mathbf{v}^i = (\mathbf{c}_t^i)_{t=0}^{T_i-1}$ is a video consisting of $T_i$ temporally overlapping clips $\mathbf{c}_t^i$ (i.e., sampled with stride 1) and $\mathbf{a}^i = (a_t^i)_{t=0}^{T_i-1}$ is the sequence of ground-truth fixations for the $i$-th video aligned with the clips. Since we use C3D to represent each clip, $\mathbf{c}_t^i$ has a fixed length of $K = 16$ frames and $t = 0$ means that the first $K$ frames are used to build the 0-th clip. The fixations $a_t^i = \{a_{t,j}^i\}_{j=1}^A$ are a set of $(x, y)$ image positions that are normalized to $[0, 1]$ in order to deal with videos of different resolutions. The number of fixations vary from frame to frame in general, but in our experiments we control it via subsampling in order to obtain for each frame a set of fixed size $A$.

Let $\mathbf{x}_t = \texttt{C3D}(\mathbf{c}_t)$ be the internal representation of C3D for an input clip $\mathbf{c}_t$. In our model we use the last convolutional layer, before the fully-connected layers. We choose a convolutional layer instead of a fully-connected layer because the latter discards spatial information, which is crucial to estimate a spatially-variant saliency map over the image.

The LSTM network (Hochreiter & Schmidhuber, 1997) is defined as follows:

$$f_t = \sigma(W_f \cdot [h_{t-1}, \mathbf{x}_t] + b_f), \qquad\qquad i_t = \sigma(W_i \cdot [h_{t-1}, \mathbf{x}_t] + b_i) \tag{1}$$

$$o_t = \sigma(W_o \cdot [h_{t-1}, \mathbf{x}_t] + b_o), \qquad\qquad \tilde{C}_t = \tanh\left(W_C \cdot [h_{t-1}, \mathbf{x}_t] + b_C\right) \tag{2}$$

$$C_t = f_t * C_{t-1} + i_t * \tilde{C}_t, \qquad\qquad h_t = o_t * \tanh\left(C_t\right) \tag{3}$$

where $f_t$, $i_t$, $o_t$, $C_t$ and $h_t$ are the forget gate, the input gate, the output gate, the memory cell, and the hidden representation, respectively. The learning parameters that need to be estimated during the training phase are $W_z$ and $b_z$, where $z \in \{f, i, o, C\}$.

The MDN (Graves, 2013; Bishop, 1994) takes its inputs from the hidden representation of the LSTM network. Since the output space is 2D (the space of image locations), we can reparametrize the model as $\{(\mu_t^c, \pi_t^c, \sigma_t^c, \rho_t^c)\}_{c=1}^C$, where $\mu_t^c$, $\pi_t^c$, $\sigma_t^c$ and $\rho_t^c$ are the 2D mean position, the weight, the 2D variance and the correlation of the $c$-th Gaussian component, respectively. The MDN is therefore defined as follows:

$$y_t = \{(\tilde{\mu}_t^c, \tilde{\pi}_t^c, \tilde{\sigma}_t^c, \tilde{\rho}_t^c)\}_{c=1}^C = W_y \cdot h_t + b_y \tag{4}$$

where $W_y$ and $b_y$ are the parameters of the linear layer and $h_t$ is the hidden representation of the LSTM network. The parameters of the GMM in Eq. 4 are normalized as follows in order to obtain a valid probability distribution:

$$\mu_t^c = \tilde{\mu}_t^c, \quad \pi_t^c = \frac{\exp(\tilde{\pi}_t^c)}{\sum_{i=1}^C \exp(\tilde{\pi}_t^i)}, \quad \sigma_t^c = \exp(\tilde{\sigma}_t^c), \quad \rho_t^c = \tanh\left(\tilde{\rho}_t^c\right). \tag{5}$$

The composition of the LSTM and the MDN results in the RMDN.

## 3.2 Training

The proposed model can be trained by optimizing the log-likelihood of the training ground truth fixations, $\mathbf{a}^i$, under the GMM. The loss function for the $i$-th video, $\mathbf{v}^i$, is defined as the negative log-likelihood of the fixations under the GMM, as follows:

$$L(\mathbf{v}^i, \mathbf{a}^i) = \sum_{t=0}^{T_i-1} \sum_{j=1}^A -\log\left(\sum_{c=1}^C \pi_t^c \mathcal{N}(a_{t,j}^i; \mu_t^c, \sigma_t^c, \rho_t^c)\right) \tag{6}$$

where $\mathcal{N}$ is the Gaussian distribution. Note that the parameters of the Gaussian components depend on the input video $\mathbf{v}_i$, but we do not make this explicit in the equation in order to keep notation simple.

The log-likelihood of the RMDN is optimized using backpropagation through time, since it is a composition of continuous functions (e.g. linear transformations and element-wise non-linearities) and the LSTM, for which we can compute the gradients. In particular, we refer to the paper of Graves (2013) for the derivation of the gradients for the MDN using the loss function of Eq. 6. In practice, due to the limited training data, we freeze the layers of the C3D network to the values pretrained by Tran et al. (2015) for action recognition. This implies that the low-level representation $\mathbf{x}_t$ is fixed. We jointly train the LSTM and MDN from randomly initialized parameters.

## 3.3 Prediction

The inference stage is straightforward by following the equations of Sec. 3.1. At a given time $t$, the clip from time $t - K + 1$ to $t$ is fed into the C3D network to produce the representation $\mathbf{x}_t$. This vector is passed to the LSTM (Eqs. 1, 2, and 3) whose hidden representation is passed to the MDN, which outputs the GMM parameters (Eqs. 4 and 5). In order to generate the final saliency map, we compute the probability of each pixel position under the GMM model. We normalize the probability map to sum up to 1 over the image pixels in order to produce a normalized saliency map.

## 3.4 Saliency for Classification

For the task of video classification, we generate a modified version of the video by using a soft-attentional mechanism: the idea is to weight each pixel value by the estimated saliency at that

position. This operation effectively down-weights regions that are deemed not salient. The intuition is that then the classifier will be able to focus on the parts of the frame which are most relevant without being distracted by the non-salient regions (see Fig. 2 in Appendix A).

At each time $t$, we extract two representations: the "context" branch is given by the C3D representation of the original clip, while the "soft attentional" branch is given by the C3D representation of the input clip weighted by the saliency map. The rationale is that the context branch considers the global evolution of the activity in the video while the soft attentional branch is focused on the most-salient local evolution of the activity. The two representations are then concatenated at the clip level and max-pooled over the video to obtain the final video-level descriptor. This video-level representation is then used as input to train the video classifier, which is a linear SVM in all our experiments.

## 4    EXPERIMENTS

In this section, we evaluate the proposed method for both saliency prediction and action recognition on two challenging datasets: Hollywood2 (Marszałek et al., 2009) and UCF101 (Soomro et al., 2012). Section 4.1 reports a quantitative analysis for the task of saliency prediction. Section 4.2 shows the results for the action recognition task in two scenarios: 1) using the same dataset that was used to train the saliency predictor and 2) applying the pretrained attentional model to a never-seen dataset and a different set of actions. We reported the implementation details in Appendix B. We invite the reader to watch the qualitative results of the proposed method in the form of a video available at `https://youtu.be/aXOwc17nx_s`.

### 4.1    SALIENCY PREDICTION

The proposed model is trained using human fixation data. Few datasets provide both human fixations and class labels, which we need for the action recognition experiment discussed in the next section. Therefore, we used the Hollywood2 dataset, which was augmented with eye tracking data by Mathe & Sminchisescu (2015). We follow the same evaluation protocol (*i.e.*, same training/test splits) of Mathe & Sminchisescu (2015) and their validation procedure to compute the final results in order to compare with their work. Mathe & Sminchisescu (2015) generate the ground truth saliency from a binary fixation map where the only non-zero values are at fixations points. The final saliency map is produced by convolving the binary map with an isotropic Gaussian filter with standard deviation $\sigma$ and then adding to it a uniform distribution with probability parameter $p$. As in Mathe & Sminchisescu (2015), the values of these two parameters are chosen from $\sigma \in \{1.5, 3.0\}$ and $p \in \{0.25, 0.5\}$ via hold-out validation. We use a validation set consisting of $20\%$ of the training set. We use the remaining $80\%$ of the training data to learn our models, and use the hold-out validation set to choose the hyperpameters of our model.

We evaluate all models on the test set, using popular metrics proposed in the literature of saliency map prediction for still images (Judd et al., 2012; Borji et al., 2013), such as Area Under the ROC Curve (AUC), Normalized Scanpath Saliency (NSS), linear Correlation Coefficient (CC) and the Similarity score (Sim). We refer to the papers of Judd et al. (2012) and Borji et al. (2013) for their detailed description.

Table 1 shows results achieved with different variants of our model and a simple baseline method, which we refer to as Trained Central Bias (TCB). The TCB model is a single GMM trained using the fixations of all the videos in the training sets. TCB predicts the same saliency map for each testing frame, thus it discards completely the temporal information and the image input. This experiment shows that all versions of our RMDN consistently outperform TCB under all metrics, even when using a smaller number of fixations per frame during training.

The different variants of RMDN in Table 1 explore the following design choices in our model: 1) the impact of using LSTM hidden units as opposed to regular RNN units (second and third row) and 2) the number of fixations per frame used for training (third and fourth row). These experiments show that LSTM (third row) is better than an RNN (second row) in terms of AUC and NSS, but in order to have better CC and Sim we need to use more fixations per frame (fourth row). This is intuitive: since the LSTM has many more parameters than the RNN, it needs more training data to be properly optimized.

The last row in Table 1 shows the results obtained by retraining our model using the full training set of Mathe & Sminchisescu (2015) instead of just the $80\%$ subset. For this case (RMDN full) we used the hyper-parameter values selected via hold out-validation for the experiment in the fourth row. This gives the best result for saliency prediction reported in our work and it is the model we used for all the subsequent experiments described below.

All of the experiments reported in Table 1 were obtained using $C = 20$ components in the GMM. We have also studied how the accuracy varies by reducing the value of $C$. For example, using the RMDN variant of row 5 in Table 1 but with $C = 10$ components (instead of 20), the performance does not change dramatically, yielding AUC= $0.8966$ and NSS= $2.4392$. On the other hand, the AUC and the NSS decrease considerably, by $1.3\%$ and $0.3$ points respectively, when using only $C = 2$ components (AUC= $0.8836$ and NSS= $2.1385$). Based on this analysis, in all our subsequent experiments we used $C = 20$ as we noticed that our approach implements automatically a sort of Occam's razor, setting the weights $\pi_c$ of many components close to zero when necessary.

We have also carried out a few side experiments and discovered that using the fully-connected features of C3D instead of the convolutional representation gives results that are at least $1.5\%$ lower in terms of AUC. Moreover, we tried to finetune the C3D network for action categorization on Hollywood2. However we did not obtain any significant improvement, confirming the findings of Tran et al. (2015): the C3D representation is already general enough to perform effectively on different action recognition tasks and fine-tuning the model on smaller-scale datasets (such as Hollywood2) does not seem beneficial. We also experimented with deep LSTMs, but we obtained an insignificant improvement of performance. For this reason and also because deep LSTMs have more parameters and are more computationally expensive to train, we chose to use a shallow one-layer LSTM. Finally, we run the ablation study where the recurrent link between time $t - 1$ and $t$ of the RMDN is removed: the results in terms of AUC are $1.2\%$ and $2.4\%$ lower with respect to the RMDN which uses RNN (second row of Table 1) and LSTM (third row of Table 1), respectively.

Table 1: Accuracy of saliency prediction for the Trained Central Bias baseline and different variants of our RMDN model in terms of AUC, NSS, CC and Similarity. Training and testing are performed on disjoint splits of the Hollywood2 dataset.

| Model | Net(#units) | Fix. per frame | AUC | NSS | CC | Sim |
|---|---|---|---|---|---|---|
| Trained Central Bias | – | 150 | 0.8725 | 1.7646 | 0.5297 | 0.4812 |
| RMDN | RNN(128) | 80 | 0.8745 | 1.9505 | 0.5495 | 0.4962 |
| RMDN | LSTM(128) | 80 | 0.8866 | 2.0155 | 0.4606 | 0.4219 |
| RMDN | LSTM(256) | 150 | 0.8986 | 2.5169 | 0.6007 | 0.5278 |
| RMDN full | LSTM(256) | 150 | **0.9037** | **2.6455** | **0.6129** | **0.5349** |

We also compared our approach to the state-of-the-art in saliency prediction from video. Table 2 includes the results of the best methods taken from the extensive analysis done in Mathe & Sminchisescu (2015). The table reports also some useful baselines, such as the central bias (CB) and the human accuracy for the task. Note that: 1) CB differs from TCB, since it does not use any training fixations; and 2) the human accuracy is computed in (Mathe & Sminchisescu, 2015) by deriving a saliency map from half of our human subjects and is evaluated with respect to fixations of the remaining ones. Furthermore, Table 2 contrasts the use of static features, motion features and their combination. The last row reports the results obtained with our RMDN model. It is interesting to see that the results obtained with a single type of features (static or motion) have an AUC lower than $0.75$, which is even lower than the one obtained by the central bias ($0.84$). Moreover, the combination reaches the best results when the central bias is combined with engineered features (SF+MF+CB). On the other hand, our method outperforms all the methods evaluated in Mathe & Sminchisescu (2015) by a large margin and our results are very close to human performance (the difference is only $3.2\%$). In addition to being the best method in Table 2, our method has several advantages: 1) it does not require any hand-engineering of spatiotemporal features, 2) it performs joint training of the LSTM and the saliency predictor, 3) it is very efficient. Specifically, although we cannot estimate the runtime for prior approaches, we believe that our method is much faster than most of the methods reported in Table 2 as these depend on features that are computationally expensive to extract. Our proposed method takes only $0.08s$ per clip for inference on GPU: $0.07s$ to compute C3D features and $0.01s$ to evaluate the RMDN.

Table 2: Saliency prediction comparison against the state-of-the-art on the Hollywood2 dataset. The top-3 best results for each set are taken from (Mathe & Sminchisescu, 2015)

| Set | Model | AUC |
|---|---|---|
| Baselines | Uniform | 0.500 |
| | Central Bias (CB) | 0.840 |
| | Trained Central Bias (TCB) | 0.872 |
| | Human | **0.936** |
| SF = Static Features | Color features (Judd et al., 2009) | 0.644 |
| | Saliency map (Oliva & Torralba, 2001) | 0.702 |
| | Horizon det. (Oliva & Torralba, 2001) | 0.741 |
| MF = Motion Features (Mathe & Sminchisescu, 2015) | Flow magnitude | 0.626 |
| | Flow bimodality | 0.637 |
| | HOG-MBH det. | 0.743 |
| Combo (Mathe & Sminchisescu, 2015) | SF (Judd et al., 2009) | 0.789 |
| | SF+MF | 0.812 |
| | SF+MF+CB | 0.871 |
| Our Method | **RMDN** | **0.904** |

## 4.2 ACTION RECOGNITION

In order to show how saliency can be used for action recognition we carried out a set of experiments covering two scenarios: 1) using the same dataset where the saliency predictor was trained (Hollywood2) and 2) using a never-seen dataset with a different set of actions (UCF101).

The results on Hollywood2 are reported in terms of mean Average Precision (mAP) as done by Mathe & Sminchisescu (2015). Table 3 shows an analysis of 1) the impact of using different feature representations as well as 2) the effect of the saliency map. As in Tran et al. (2015), we experimented with different features, namely CONV5 and FC6, which correspond to the fifth convolutional layer and the first fully-connected representation of C3D, respectively. We also tested two ways to use the saliency maps, called in the second column: "feature" and "clip". In the feature mode (first row, experiments (2-5)), the convolutional representation is multiplied by the saliency map, after resizing it accordingly. In other words, the saliency weights directly the feature representation, similarly to the work of Sharma et al. (2016). In the clip mode (second and third row, experiments (2-5)), we adopted the model presented in Sec. 3.4, where the saliency maps are used to weight the input video pixels.

The third column of Table 3 (experiment (1)) reports the results using only the original video as input to C3D (referred to as context in Fig. 2). Experiment (2) uses the ground truth saliency maps as soft attention to weight the input of C3D, while in experiment (3) this vector is concatenated with the context features. The last two columns (experiment (4) and (5)) represent the same setup, but in this case we use the saliency maps predicted by our model instead of the ground truth.

Table 3 shows that the results of CONV5 and FC6 are very close when considering the original video (experiment (1)). The table also shows that the feature mode has lower performance compared to the clip mode (experiment (2)). Moreover, the concatenation (experiment (3)) is effective only when visual attention is used to weight pixels rather than features. Based on the poor performance of the features mode, we decided to experiment only with the clip mode in our study with predicted saliency (experiment (4) and (5)). Also, we decided to use FC6 features for the rest of the paper because the representation is more compact and therefore allows to train the classifiers more quickly. We can notice that even in the case of predicted saliency, the concatenation of FC6 context features and those obtained by weighting the input video with soft-attention (experiment (5)) produces a significant improvement over the original CONV5/FC6 features without attention. Furthermore, a pleasant surprise is represented by the the small difference in results between using the predicted saliency (experiment (5)) versus the ground truth maps (experiment (3)): only $0.27\%$ for FC6 (last row).

The SVM model complexity for experiment (3) in Table 3 is twice as large as the complexity for (1) and (2) since the feature dimensionality is doubled by construction. The same applies when comparing (5) against (1) and (4). In order to have a more fair comparison, we added PCA dimensionality reduction to experiment (5) in order to match the same feature dimensionality as (1) (and (4)). Although the validation accuracies are very similar, the testing mAP drops from $54.85\%$ of experiment (5) to $51.82\%$ of the PCA experiment. This is not surprising, since the extra dimensions

Table 3: Action categorization results in terms of mAP on the Hollywod2 dataset. Analysis of different ways to use the saliency map and comparison between using the ground truth saliency maps versus those predicted by our model.

| Saliency | | | | Ground Truth | | Predicted | |
|---|---|---|---|---|---|---|---|
| Input | Saliency Use | (1) Original | (2) Weighted | (3) Concat. $(1,2)$ | (4) Weighted | (5) Concat. $(1,4)$ |
| CONV5 | Feature | 46.08% | 40.76% | 45.62% | N/A | N/A |
| CONV5 | Clip | 46.08% | 44.89% | 55.49% | 39.42% | 53.41% |
| FC6 | Clip | 47.00% | 41.78% | 55.12% | 39.00% | 54.85% |

Table 4: Recognition results in terms of mAP for the Hollywood2 dataset. The proposed method (RMDN) is compared to the approaches reported by Mathe & Sminchisescu (2015) (named as central bias and saliency sampling). Note that Mathe & Sminchisescu (2015) and RDMN do not use the same video classification model.

| | Ground Truth | | Predicted | | |
|---|---|---|---|---|---|
| Class | SalSampling | **our RMDN** | Central Bias | SalSampling | **our RMDN** |
| AnswerPhone | 28.1% | 21.8% | 23.3% | 23.7% | 29.8% |
| DriveCar | 94.8% | 89.2% | 92.4% | 92.8% | 91.6% |
| Eat | 67.3% | 59.4% | 58.6% | 70.0% | 49.1% |
| FightPerson | 80.6% | 80.9% | 76.3% | 76.1% | 79.2% |
| GetOutCar | 55.1% | 78.0% | 49.6% | 54.9% | 76.9% |
| HandShake | 27.6% | 58.6% | 26.5% | 27.9% | 47.0% |
| HugPerson | 37.8% | 27.5% | 34.6% | 39.5% | 37.9% |
| Kiss | 66.4% | 52.2% | 62.1% | 61.3% | 51.0% |
| Run | 85.7% | 85.5% | 77.8% | 82.2% | 83.2% |
| SitDown | 62.5% | 31.8% | 62.1% | 69.0% | 31.4% |
| SitUp | 30.7% | 38.0% | 20.9% | 29.7% | 39.7% |
| StandUp | 58.2% | 37.8% | 61.3% | 63.9% | 41.3% |
| Mean | 57.9% | 55.1% | 53.7% | 57.6% | 54.8% |

provided by the use of the saliency map are not redundant with respect to the context representation (1). Therefore, concatenation seems to be an effective way to make use of the saliency map.

Table 4 compares our action categorization results with those presented in Mathe & Sminchisescu (2015). As we did before, we separate experiments that use the ground truth maps and those that use predicted saliency. The results of Table 4 show that the performance our method (second and fifth column) is around 2% lower than Mathe & Sminchisescu (2015). However this is most likely explained by the differences in the type of features and classifier, and not by the differences in saliency map prediction methods. Indeed, we already established in Table 2 that our proposed saliency map predictor is more accurate than the one proposed in Mathe & Sminchisescu (2015). On the other hand, Mathe & Sminchisescu (2015) use a combination of many different features and a kernel chi-square SVM, while our method uses C3D features with a simple linear SVM classifier. Adding more non-linearities, especially for the concatenation experiment, would probably help. But we consider the experimentation with different types of action recognition features and classifiers out of the scope of this paper.

Finally, we perform an experiment to assess the generalization abilities of the learned saliency model to a different dataset, with classes and videos that have not been seen during its training. To this end, we used the attentional model trained on the Hollywood2 dataset to extract saliency maps on the UCF101 dataset. As saliency ground truth is not available for UCF101, we evaluate performance in terms of action recognition accuracy using the evaluation protocol and splits by Soomro et al. (2012). Table 5 summarizes the results. The proposed method (C3D + RMDN, eight row) corresponds to the concatenation of the original C3D descriptor and the C3D descriptor with the input weighted by the saliency map, as was done in the Hollywood2 experiments. We compare our method with the results obtained using the C3D descriptor computed from the context only (seventh row) and other state-of-the-art methods (first row through sixth row). A linear SVM trained on C3D features computed from the context already outperforms most of the other methods (first row to forth row). But training the linear SVM on a concatenation of context C3D features and those obtained by reweighting the video

input with the RMDN saliency maps (seventh row) leads to a further improvement of $1.1\%$. This is an impressive result since the RMDN was trained on the separate and small Hollywood2 dataset.

Since we noticed that the saliency maps of RMDN for UCF101 tend to be highly peaked around a single location in each frame, we added the trained central bias (already analyzed in Table 1). This has the effect of diffusing the saliency map with the central bias, thus enlarging the area of attention used by the recognition system. The result of this experiment, which is reported in the last row of Table 5, further improves the accuracy by $1.3\%$.

Table 5: Action categorization results in terms of 3-fold accuracy on the UCF101 dataset.

| Method | Accuracy |
|---|---|
| Imagenet + linear SVM | 68.8% |
| iDT (Wang & Schmid, 2013) + BoW + linear SVM | 76.2% |
| Spatial stream network (Simonyan & Zisserman, 2014) | 72.6% |
| LSTM composite model (Srivastava et al., 2015) | 75.8% |
| scLSTM (Song et al., 2016) | 84.0% |
| LSTM (optical flow, images) (Yue-Hei Ng et al., 2015) | 88.6% |
| C3D + linear SVM | 80.4% |
| C3D + RMDN + linear SVM | 81.5% |
| C3D + RMDN + TCB + linear SVM | 82.8% |

## 5 CONCLUSIONS

In this paper, we proposed a recurrent mixture density network for spatiotemporal visual attention. We showed that our model outperforms state-of-the-art methods for saliency prediction in videos. We have also shown that the saliency maps generated by our model can be leveraged to improve action categorization using a very simple procedure. This suggests that saliency can enrich the original video representation. The runtime overhead to estimate the saliency map is very small: only 0.01s added to the feature extraction time of 0.07s.

As future work, we plan to close the gap between RMDN and action recognition with a joint network. The idea is to have as output of the model both the saliency map at each time and the class of the action for the entire video. This can be combined with the idea of using the saliency map estimated at the previous time to weight the input for the current time. Putting together these two ideas in a single network would result in a joint model for saliency prediction and action recognition.

## 6 ACKNOWLEDGMENTS

We thank Du Tran for helpful discussion about the code of the C3D network and its usage. We are grateful to Stefan Mathe for explaining the format of the eyetracking data and the protocol of the Hollywood2 experiment. This work was funded in part by NSF award CNS-1205521. We gratefully acknowledge NVIDIA for the donation of GPUs used for portions of this work.

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

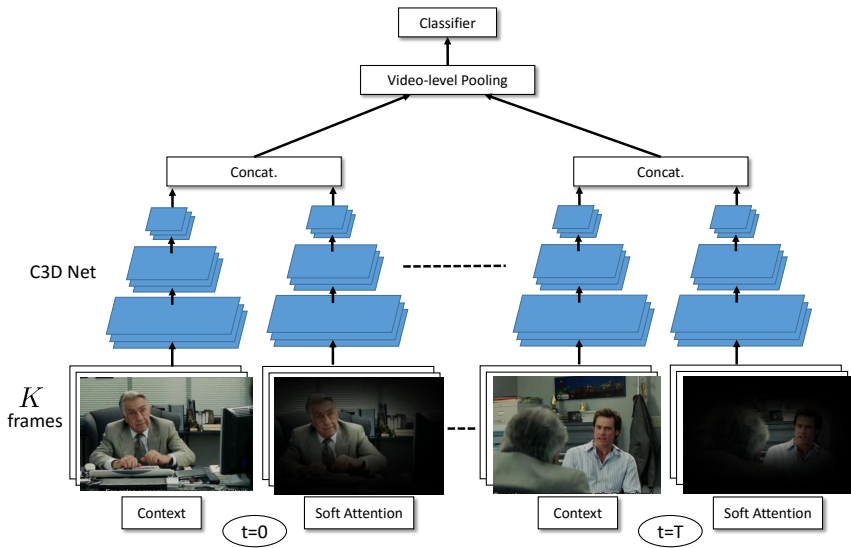

Figure 2: Model for action recognition. The original clip of $K$ frames is fed into a 3D convolutional network. The same clip is then weighted by the predicted saliency map estimated by our RMDN and then fed into the 3D convolutional network. The final clip-level representation is then concatenated. All the clips of a video are merged using pooling and then a linear classifier can be trained.

## A  SALIENCY FOR CLASSIFICATION

The proposed model for recognition is presented in Fig. 2. At each time $t$, we extract two representations: the context branch is given by the C3D representation of the original clip, while the soft attentional branch is given by the C3D representation of the input clip weighted by the saliency map. The two representations are then concatenated at the clip level and max-pooled over the video to obtain the final video-level descriptor. This video-level representation is then used as input to train the video classifier which is a linear SVM in our experiments.

In our experiments, we also evaluated the option of weighting the convolutional feature map $\mathbf{x}_t$ instead of the input, as for example done by Sharma et al. (2016). However, we will see that soft-masking the input gives higher accuracy, probably because applying C3D's non-linear transformation after the soft-weighting produces a representation that is less redundant with the original (non-masked) C3D representation.

## B  IMPLEMENTATION DETAILS

We used the pretrained C3D network (Tran et al., 2015) as feature representation which is the input of the LSTM network. The convolutional layer before the fully-connected layers is used for saliency prediction, while the last fully-connected layer before the softmax is used for classification, since Tran et al. (2015) showed to obtain the best performance.

The training of the RMDN is performed using RMSprop with adaptive learning rate and gradient clipping. We start from a learning rate of $0.0003$ and after 8 epochs it is reduced at each epoch with a decay factor of $0.95$. The gradient is clipped with a threshold of 20. Dropout with a ratio of $0.5$ is applied only on the hidden layer of the LSTM network before the MDN. We trained for 40 epochs, but training is stopped if there is no significative improvement of the loss. During training, temporal data augmentation is performed by clipping the videos to shorter videos of length 65 frames (which corresponds to 50 C3D descriptors since it needs a buffer of 16 frames for the first descriptor). The number of components of the GMM $C$ is fixed to 20 for all the experiments. All the experiments were carried out using an NVIDIA Tesla K40 card.

After extracting the saliency maps and the feature representations on GPU, our recognition experiments were performed on CPU using a linear SVM. In order to compute the video-level representation, we performed max pooling of the clip-level representations of the video. For all the experiments, we used $20\%$ of the training data as validation set to find the regularization parameter of the SVM. We searched the parameter space on a grid between $10^{-9}$ to $10^3$ with a step of $10^{\frac{1}{2}}$. Finally, we retrain the SVM on all the training set (including the validation set) using the best cross-validated parameter.

