# Peer review of "Recurrent Mixture Density Network for Spatiotemporal Visual Attention"

_ICLR 2017 — accepted_

[Official Review · AnonReviewer3 · rating 6 · confidence 4 · 15 Dec 2016]
**No Title**

This paper proposes a new method for estimating visual attention in videos. The input clip is first processed by a convnet (in particular, C3D) to extract visual features. The visual features are then passed to LSTM. The hidden state at each time step in LSTM is used to generate the parameters in a Gaussian mixture model. Finally, the visual attention map is generated from the Gaussian mixture model.

Overall, the idea in this paper is reasonable and the paper is well written. RNN/LSTM has been used in lots of vision problem where the outputs are discrete sequences, there has not been much work on using RNN/LSTM for problems where the output is continuous like in this paper.

The experimental results have demonstrated the effectiveness of the proposed approach. In particular, it outperforms other state-of-the-art on the saliency prediction task on the Hollywood2 datasets. It also shows improvement over baselines (e.g. C3D + SVM) on the action recognition task.

My only "gripe" of this paper is that this paper is missing some important baseline comparisons. In particular, it does not seem to show how the "recurrent" part help the overall performance. Although Table 2 shows RMDN outperforms other state-of-the-art, it might be due to the fact that it uses strong C3D features (while other methods in Table 2 use traditional handcrafted features). Since saliency prediction is essentially a dense image labeling problem (similar to semantic segmentation). For dense image labeling, there has been lots of methods proposed in the past two years, e.g. fully convolution neural network (FCN) or deconvnet. A straightforward baseline is to simply take FCN and apply it on each frame. If the proposed method still outperforms this baseline, we can know that the "recurrent" part really helps.

[Official Review · AnonReviewer1 · rating 7 · confidence 4 · 16 Dec 2016]
**Interesting paper, some evaluation issues**

The authors formulate a recurrent deep neural network to predict human fixation locations in videos as a mixture of Gaussians. They train the model using maximum likelihood with actual fixation data. Apart from evaluating how good the model performs at predicting fixations, they combine the saliency predictions with the C3D features for action recognition.

quality: I am missing a more thorough evaluation of the fixation prediction performance. The center bias performance in Table 1 differs significantly from the on in Table 2. All the state-of-the-art models reported in Table 2 have a performance worse than the center bias performance reported in Table 1. Is there really no other model better than the center bias? Additionally I am missing details on how central bias and human performance are modelled. Is human performance cross-validated?

You claim that your "results are very close to human performance (the difference is only 3.2%). This difference is actually larger than the difference between Central Bias and your model reported in Table 1. Apart from this, it is dangerous to compare AUC performance differences due to e.g. saturation issues.

clarity: the explanation for Table 3 is a bit confusing, also it is not clear why the CONV5 and the FC6 models differ in how the saliency map is used. At least one should also evaluate the CONV5 model when multiplying the input with the saliency map to see how much of the difference comes from the different ways to use the saliency map and how much from the different features.

Other issues:

You cite Kümmerer et. al 2015 as a model which "learns ... indirectly rather than from explicit information of where humans look", however the their model has been trained on fixation data using maximum-likelihood.

Apart from these issues, I think the paper make a very interesting contribution to spatio-temporal fixation prediction. If the evaluation issues given above are sorted out, I will happily improve my rating.

[Official Review · AnonReviewer2 · rating 6 · confidence 4 · 16 Dec 2016]

This work proposes to a spatiotemporal saliency network that is able to mimic human fixation patterns,
thus helping to prune irrelevant information from the video and improve action recognition.

The work is interesting and has shown state-of-the-art results on predicting human attention on action videos.
It has also shown promise for helping action clip classification.

The paper would benefit from a discussion on the role of context in attention.
For instance, if context is important, and people give attention to context, why is it not incorporated automatically in your model?

One weak point is the action recognition section, where the comparison between the two (1)(2) and (3) seems unfair.
The attention weighted feature maps in fact reduce the classification performance, and only improve performance when doubling the feature and associated model complexity by concatenating the weighted maps with the original features.

Is there a way to combine the context and attention without concatenation?
The rational for concatenating the features extracted from the original clip,
and the features extracted from the saliency weighted clip seems to contradict the initial hypothesis that `eliminating or down-weighting pixels that are not important' will improve performance.

The authors should also mention the current state-of-the-art results in Table 4, for comparison.

# Other comments:
# Abstract
- Typo: `mixed with irrelevant ...'
``Time consistency in videos ... expands the temporal domain from few frames to seconds'' - These two points are not clear, probably need a re-write.

# Contributions
- 1) `The model can be trained without having to engineer spatiotemporal features' - you would need to collect training data from humans though.. 

# Section 3.1
The number of fixation points is controlled to be fixed for each frame - how is this done?

In practice we freeze the layers of the C3D network to values pretrained by Tran etal.
What happens when you allow gradients to flow back to the C3D layers?
Is it not better to allow the features to be best tuned for the final task?

The precise way in which the features are concatenated needs to be clarified in section 3.4.

Minor typo:
`we added them trained central bias'

[Final Decision · Program Chairs · 06 Feb 2017]
**ICLR committee final decision**

The paper describes a model for video saliency prediction using a combination of spatio-temporal ConvNet features and LSTM. The proposed method outperforms the state of the art on the saliency prediction task and is shown to improve the performance of a baseline action classification model.